# Mediterranean Diet, Psychological Adjustment and Health Perception in University Students: The Mediating Effect of Healthy and Unhealthy Food Groups

**DOI:** 10.3390/nu13113769

**Published:** 2021-10-25

**Authors:** Mercedes Vélez-Toral, Zaira Morales-Domínguez, María del Carmen Granado-Alcón, Diego Díaz-Milanés, Montserrat Andrés-Villas

**Affiliations:** 1Department of Social, Developmental and Educational Psychology, University of Huelva, 21007 Huelva, Spain; maria.velez@dpee.uhu.es (M.V.-T.); montserrat.andres@dpsi.uhu.es (M.A.-V.); 2Department of Clinical and Experimental Psychology, University of Huelva, 21007 Huelva, Spain; zaira.morales@dpsi.uhu.es; 3Department of Psychology, Universidad Loyola Andalucía, 41704 Sevilla, Spain

**Keywords:** Mediterranean diet, university students, mediation, health perception, psychological adjustment

## Abstract

This study aims to identify the relationships between eating habits and psychological adjustment and health perception, and to analyze potential mediating role of healthy and unhealthy foods in the relationship between adherence to the Mediterranean diet (MedDiet) and the psychological constructs and health perception. The sample was selected through stratified random cluster sampling and was composed of 788 university students. The participants responded to a MedDiet adherence screener and food consumption inventory to assess the eating habits, instruments measuring self-esteem, life satisfaction, curiosity and sense of coherence to assess the psychological adjustment, and single item measuring perceived health. The results revealed 41.9% of the participants had a high consumption of vegetables and 85.1% a low consumption of energy drinks, while 29.9% showed a high adherence to the MedDiet which was positively associated to each psychological variable and healthy foods and negatively with unhealthy foods. In conclusion, a higher adherence to the MedDiet, and the consumption of fruits and vegetables is related to higher psychological adjustment and health perception. However, the relationships between MedDiet and the psychological variables and health perception were fully or partially explained because of the consumption of healthy and unhealthy foods.

## 1. Introduction

Diet is a key element in public health [1]. The WHO describes it as one of the relevant aspects to be addressed within its Strategic Intervention Plan 2019–2023 [2]. Research studies based on the analysis of health-promoting indicators provided by the adoption of a healthy diet suggest a solid biological basis for the benefits generated by the Mediterranean diet (MedDiet) [3], and define the latter as the frequent intake of vegetables, legumes, fruits, nuts, cereals, olive oil, and fish, as well as moderate or low consumption of dairy products, meats, eggs or saturated fat [4,5,6]. This consumption pattern is associated with optimizing the maturational development of individuals throughout the life cycle and is positively associated with an increase in their life expectancy [7]. On the other hand, evidence indicates that the diet of university students is characterized by being hypercaloric and unstable [8,9], associated with high levels of consumption of alcoholic and sugary beverages and processed foods (high in “fat, sugar and sodium” and low in fiber) and low levels of fruit and vegetable consumption [10] as well as insufficient consumption of olive oil, whole grains and nuts [10], a pattern that is particularly visible at exam time [11]. Thus, it seems that university students, due to the characteristics of the important life stage in which they find themselves, would be at greater risk since, for many, the start of university studies entails moving away from their family home and having to take responsibility for different aspects of their lives, including the type of food they buy and consume. This makes this stage a critical time for the development of their eating habits [12].

Some of the indicators that have been associated with this growing trend of abandoning MedDiet for a less healthy diet more typical of Western societies are: being male, living outside the family home, peer group pressure (which influences not only diet but also the adoption of other lifestyle behaviors such as substance use), having a low socioeconomic status, and engaging in low levels of physical activity [13,14,15]. Additionally, other factors have been revealed as barriers in the university population such as the prices food and equipment, culinary knowledge and the cooking skill, among others [16,17].

Moreover, aside from studies on the impact of MedDiet, others have analyzed the role of food groups or what is known as a healthy diet, examining the possible connection between food and well-being. The findings suggest that a healthy diet can improve the perception of well-being and health, physical health, and psychological adjustment [18,19].

The perception of health and quality of life is a multifactorial indicator that refers to the subjective evaluation that each individual constructs concerning their own physical and mental health and sense of well-being [13]. Various research studies have found that perceived health and well-being is linked to the lifestyle maintained by the individual. Thus, an unhealthy lifestyle and a low perception of health and well-being are associated with cardiovascular disease in all population groups, including young adults [20,21]. Analysis of the indicators implicated in this relationship highlights the importance of diet as a relevant factor, particularly fruit and vegetable consumption [22]. In addition, it has been identified that healthy eating is related to health perception, such that compliance with a greater number of healthy eating criteria is associated with better perceived health [23,24].

Concerning psychological adjustment, it is important to pay attention to self-esteem [25], which Rosenberg understands as the feeling towards oneself constructed by the valuation of one’s characteristics [26]. It has been demonstrated that a more positive body image is associated with better self-esteem [27,28,29,30,31], and it has even been shown that self-esteem has a predictive value in relation to body dissatisfaction [32]. Regarding the role of food, in the study by Castañeda et al. [33], it was observed that although most university students showed a high level of self-esteem, only a minority claimed to have excellent eating habits in terms of fruit and vegetable intake. A possible explanation for this finding could lie in the role played by self-esteem in social adjustment [25], so food would not only exert its influence due to its nutritional value but also due to the social meanings attached to it [34], which are highly important at this stage of life.

Another variable that has been considered when discussing psychological adjustment is the sense of coherence (SOC). As conceptualized by Antonovsky [35]. This is one of the central concepts of the salutogenic model. It refers to an overall personal disposition that expresses a person’s capacity to handle demanding situations. A strong SOC would facilitate successful management of stressful daily events and successful adaptation [35], even under adverse conditions [36].

It appears that, in general, the higher the SOC, the greater the tendency to adopt healthy behaviors at any stage of life. Several studies show that people with a strong SOC show eating patterns that follow more closely the recommended guidelines and make choices more in line with a healthy lifestyle than those with a weaker SOC [37,38,39]. People with a high SOC are less likely to be smokers or sedentary, and more likely to consume fruits, vegetables, and fiber than those with a low SOC; in addition, the former were 20% less likely to die from any cause compared with the latter [40,41]. In adolescence, a high sense of SOC has been associated with adopting a healthy lifestyle [42]. In contrast, low SOC has been linked to unhealthy habits such as skipping breakfast, having positive attitudes towards drugs, or consuming alcohol [43]. Furthermore, SOC has shown positive associations with health and well-being in adolescents from different countries [44], while research over the years has also shown associations between SOC and other aspects of health. Previous studies have found a positive correlation between SOC and certain positive indicators of psychological health such as well-being, self-esteem, life satisfaction, or quality of life. At the same time, having a low SOC has been associated with negative psychological symptoms such as anxiety, depression, burnout, anger, demoralization, hostility, hopelessness, perceived stressors, and PTSD. It seems that people with high SOC levels manage stressful situations more efficiently than those with low SOC. In a study with the Spanish adolescent population, García-Moya, Moreno, and Rivera [45] found that higher SOC was related to better perceived health, lower frequency of somatic and psychological complaints, and higher quality of life and satisfaction with life (SWL). The effect of SOC on the relationships between these variables was moderate to large for SWL, quality of life, and frequency of psychological symptoms.

Another variable related to psychological adjustment is curiosity, understood as the tendency to seek novel, complex, and challenging interactions with the world [46,47]. The limited evidence in this regard indicates that as a state, curiosity facilitates the coordination of physiological states that are associated with concentration and action-oriented focus [48,49], in addition to increasing motivation to acquire new skills and knowledge [50], which could facilitate college students’ greater willingness to acquire the skills needed to eat healthily. As a trait, curiosity is positively associated with higher levels of well-being and life satisfaction [51,52,53] and is negatively associated with depression [54,55].

In view of the above, and given the relevance of food for well-being (whether this is the Mediterranean diet or food groups), and the relationships identified with psychological variables in the university population, the objectives of this study were as follows: (i) to describe the eating habits and adherence to MedDiet, psychological constructs associated with better mental health (self-esteem, SWL, SOC, and curiosity), and perceived health in university students; (ii) to analyze the possible food groups consumed by university students, (iii) to evaluate the relationship between eating and the aforementioned psychological constructs and perceived health, as well as between each of these variables in university students, and (iv) to test for the existence of mediators (“healthy foods” and “unhealthy foods”) in the relationship between adherence to MedDiet and the psychological constructs and perceived health.

## 2. Materials and Methods

### 2.1. Study Design

Stratified random cluster sampling was used to recruit the participants. The strata (with proportional allocation) were the areas of knowledge. As clusters, first and third-year subjects were randomly selected until completing the quota established for each area of knowledge. The inclusion criteria to participate in the study were to be enrolled in a degree program at the University of Huelva and give explicit consent for processing their data. The exclusion criteria were Erasmus students, students in the practical classroom, students who had previously participated in the survey, and minors (<18 years of age).

### 2.2. Variables and Instruments

Two instruments were used to assess dietary habits: (i) The Mediterranean Diet Adherence Screener (MEDAS) [56] to assess adherence to the MedDiet. This questionnaire has been adapted in different countries showing reasonable construct validity [57,58,59,60] and employed in the Spanish university population for the study of habits related to both physical and mental health [61,62,63]. The questionnaire is composed of 14 items that can score 0 (Does not meet criteria) or 1 (Meets criteria); 12 of the items refer to the frequency of consumption of certain foods (fruits, vegetables, olive oil, animal fats, red meat, nuts, industrial confectionery, fizzy drinks, wine, fish and seafood, and soft drinks) while two focus on dietary habits/characteristics consistent with a traditional MedDiet, such as preferential consumption of white over red meat and frequency of consumption related to pasta/rice/vegetables cooked with a sauce of tomato, onion, garlic, leeks sauteed in olive oil (Sofrito). The total score corresponds to the sum of each item, obtaining a final score of 0 to 14 points. Higher scores on the questionnaire indicate greater adherence to the MedDiet, with scores greater than or equal to 9 taken to indicate high adherence and scores below 9 considered low adherence.

(ii) A food inventory was included to assess the frequency of food consumption. This was extracted from Spain’s Health Behavior in School-aged Children (HBSC) study [64]. The inventory comprises six items that refer to the weekly consumption of fruits, crisps, and salty snacks, vegetables, sweets, energy drinks, and soft drinks or drinks containing sugar. Each item has 7 response options, where 1 is “Never” and 7 is “Every day, more than once”.

Concerning the psychological constructs included in the study:

Life satisfaction was measured through the Satisfaction With Life Scale (SWLS) or, in Spanish, Escala de Satisfacción con la Vida de Diener [65] in its adaptation to Spanish by Vázquez, Duque and Hervás [66]. The scale comprises 5 items with 7 Likert-type response options, where 1 is “totally disagree” and 7 is “totally agree”, to assess subjective well-being in different facets of life. This instrument has shown good validity and reliability, with alpha values of 0.79 to 0.89 [67], demonstrating its unidimensionality and adequate psychometric properties in the Spanish-speaking population [68,69] and specifically in university students [70]. The total score was calculated through the sum of the scores of each item, obtaining a range of 7 to 35 points. The scale showed good reliability, as estimated by a Cronbach’s α = 0.84.

Curiosity was assessed by the Curiosity and Exploration Inventory (CEI-II) [71]. This instrument comprises 10 items with 5 Likert-type response options, where 1 indicates “Very little or not at all” and 5 “Very much”. The internal consistency of this scale has ranged in previous studies between 0.77 and 0.90 [71,72,73,74]. Stretching, the drive to seek out new information and experiences, and Embracing, the readiness to welcome the unique, uncertain, and unpredictable aspect of everyday life, are the two domains that constitute this inventory. However, both the authors themselves [71] and subsequent studies [74] have pointed out the appropriateness of using the full-scale score as a single factor. The version of the instrument used was extracted from the HBSC study in Spain [64]. The reliability of the instrument in the present study, estimated by Cronbach’s alpha, was α = 0.89.

Self-esteem was measured with the Rosenberg Self-Esteem Scale [26]. This is a self-administered scale composed of 10 items that assess feelings of respect and acceptance about oneself through a Likert-type scale. Since different studies have used versions of this scale with a greater number of response options (from 5 to 11) showing adequate validity and reliability parameters for all versions [75,76], the version used in this present study contained 5 response options, with 1 indicating “strongly disagree” and 5 “strongly agree.” The scale has been shown to have acceptable reliability parameters with test-retest correlations between 0.82 and 0.88 and a Cronbach’s alpha between 0.77 and 0.88 [77], along with a factor structure that allows the use of its global score as an indicator of self-esteem in Spanish university students [78]. The scale’s total score was used, through the sum of the items, obtaining a range of 10 to 50 points. This instrument showed good reliability, estimated by a Cronbach’s α of 0.86.

The sense of coherence was measured through the Sense of Coherence Scale (SOC) created by Antonovsky [35]. The Spanish translation of this instrument was taken from the HBSC study in Spain [64]. The brief version was used, composed of 13 items with 7 Likert-type response options that study the frequency with which the person lives certain experiences. The scale’s total score ranges from 13 to 91 points and can be used as a single dimension or divided into three factors: meaningfulness, comprehensibility, and manageability. Numerous studies have provided evidence of the validity of this scale [79] and adequate internal consistency with values between 0.70 and 0.92 [80]. For this study, the global score was used as a single factor whose reliability, estimated by Cronbach’s Alpha, was α = 0.80.

Finally, a perceived health item [81] was included with four response options: excellent, good, passable, and poor.

### 2.3. Procedure

The present research is part of the Health Behavior in University (HBU) study, which is based on a cross-sectional survey design. The data used in this study were those collected during the 2018/2019 academic year.

Before collecting these data, a list of the different degree programs was obtained, and the approximate number of students enrolled in each subject was estimated using data from the previous year. Subsequently, quotas were established according to the area of knowledge, and a random selection procedure of the degrees and subjects per quota was carried out to administer the questionnaire.

The lecturing staff responsible were then contacted. The location and administration time were agreed upon after informing them of the purpose of the study and the duration of data collection. Data collection was then carried out in the classroom during class time, indicating to the students that their participation was voluntary and not related to the subject. They were offered the questionnaire, together with the informed consent form. Finally, the data were computerized by the various researchers under the same digitalization protocol.

### 2.4. Statistical Analysis

Initially, a descriptive analysis was conducted by calculating the mean and standard deviation for scale variables and the frequency and percentage for nominal and ordinal variables.

Principal component analysis (PCA) was employed to reduce the dimensionality of the food inventory data. Prior to this, the Kaiser-Meyer-Olkin (KMO) test of sample adequacy and Bartlett’s test of sphericity were performed. Next, the communalities were studied, and the most parsimonious and reliable factorial solution was evaluated through a parallel analysis (PA), on which a Promax rotation was applied. After this, Carmines Zeller’s theta coefficients were calculated to evaluate the scale’s internal consistency.

Pearson correlation coefficients and one-factor ANOVA tests assessed the relationship between adherence to the MedDiet and PCA components and psychological constructs and perceived health.

Finally, a set of analyses was conducted to assess the mediating role of the PCA components in the statistically significant effects obtained for the psychological constructs in the bivariate analyses and adherence to MedDiet. For this purpose, a bias-corrected bootstrap-based indirect effects analysis was conducted using the PROCESS macro for SPSS [82], following the four-step process developed by Baron and Kenny [83]. The first step evaluates the regression coefficient of the explanatory variable on the explained variable, i.e., its total effect (*path c*). Step two consists of evaluating the regression coefficient of the explanatory variable on the mediating variable (*path a*), and the third step calculates the regression coefficient of the mediating variable on the explained variable (*path b*). In the fourth and last step, the relationship between the explanatory and mediating variable and the explained variable is evaluated, extracting the direct effect of the explanatory variable (*c − b = c’*). The mediation is total if the relationship is statistically significant in the first three steps but not in the fourth. On the other hand, if the correlation decreases despite being statistically significant in the fourth step, this is taken to indicate partial mediation. In both cases, the point estimate (ab) was considered to be significant when the confidence interval did not contain zero.

To apply this process, 5000 bootstrap resamplings were carried out to determine an error correction at a 95% interval [84]. The coefficients included in the diagrams refer to standardized coefficients.

A statistical significance of less than 5% (*p* < 0.05) was used for all cases. Statistical analyses were conducted using the statistical package IBM SPSS Statistics, version 23.0 (IBM, Armonk, NY, USA) [85], with the support of Microsoft Excel software for complementary calculations.

### 2.5. Ethical Issues

The present investigation has the approval of the Research Ethics Committee of Huelva Centers (CEI) of the Junta de Andalucía with reference code 0846-N-19/P1027/19. Furthermore, it has followed the Declaration of Helsinki [86] in its design and implementation. Participation in the study was completely voluntary. The participants provided explicit permission through informed consent for the personal use and treatment of the data, following current legislation on personal data protection. The data were stored in an anonymous database, with the assignment of a registration number so that it was not possible to identify the participants. All data, both in physical and digital format, were kept by the research team.

## 3. Results

### 3.1. Characteristics of the Sample

A total of 970 students agreed to complete the instrument, of whom 31 did not adequately complete the informed consent form, 59 were minors, or did not provide data on age or were identified as outliers for the university population (extreme values greater than 26 years according to the stem and leaf graph), and 92 did not fully complete all of the instruments or items used in the study. Thus, the final study sample was composed of 788 students, of whom 75% were female and 25% were male, with an age range of 18 to 26 years (M = 20.70; SD = 2.17).

The participants in the present study belonged to the following areas of knowledge: Arts and Humanities (6% of the sample), Engineering and Architecture (1.8%), Natural Sciences (2.4%), Health Sciences (41.4%), and, finally, Social and Legal Sciences, which accounted for almost half of the sample (48.5%).

### 3.2. Descriptive Analysis

In the case of diet-related variables, moderate adherence to the MedDiet was observed in the population studied with a mean score of 7.42 (SD = 2.06), where the 29.9% of participants obtained scores above 9, indicating that they had a high level of adherence to MedDiet.

On the other hand, the food inventory showed very low consumption of energy drinks, low consumption of soft drinks and sweets, medium-low consumption of chips and salty snacks, medium consumption of fruits and medium-high of vegetables (Table 1).

Regarding the psychological constructs measured the average score for life satisfaction was 25.07 (SD = 5.82), 33.62 (SD = 7.37) for curiosity, 37.68 (SD = 7.07) for self-esteem and 57.72 (SD = 11.9) for SOC. Moreover, it was observed that most participants indicated their health as good (72%), followed by passable (13.6%), excellent (13.1%), and poor (1.4%).

### 3.3. Principal Component Analysis

The possibility of grouping the inventory of eating habits used was analyzed through Principal Component Analysis (PCA). The results of the Kaiser-Meyer-Olkin test (KMO =. 659) and Bartlett’s sphericity test (*χ*^2^(15) = 596.586; *p* < 0.001) showed that the responses to the scale could be considered adequate for further analyses.

A parallel analysis was conducted to determine the number of components to be extracted through a statistical criterion of eigenvalue comparison. For this purpose, one thousand random simulations of the data matrix were used, performing the PCA on each. Then, the 95th percentile of the eigenvalues of each component was calculated, comparing the eigenvalue of each component of the observed matrix with its respective 95th percentile of the random values generated. Only the components whose observed value was greater than the random result were retained. The third observed component obtained a value lower than the 95th percentile of the third random component of the thousand matrices (third observed eigenvalue =0.95 < = 1.05 third parallel eigenvalue). Thus, the first two observed components have a cumulative explained variance of 54.79% (Table 2).

An oblique rotation (Promax method) was then applied to these components. Table 3 shows the factor loadings of each item after eliminating those below 0.500.

The communalities indicated a good representation of the information in the extracted components, Component 1 (“unhealthy foods”) and Component 2 (“healthy foods”), except for the item referring to the consumption of energy drinks (0.26).

The reliability of the reduction in dimensions, calculated using Carmines and Zeller’s theta coefficient, indicating the suitability for dimension reductions using PCA, was 0.629.

### 3.4. Bivariate Analysis

Total MEDAS scores and consumption of “healthy foods” (fruits and vegetables) showed statistically significant positive correlations with SWL, curiosity, self-esteem, SOC, and with each other, while they correlated inversely with consumption of unhealthy foods (Potato chips and sweet-salty snacks and soft drinks). Likewise, consumption of unhealthy foods showed a statistically significant correlation with SWL and a marginal correlation with self-esteem (Table 4).

Each food-related variable was analyzed according to health perception, finding statistically significant differences in the MEDAS scores (F (3, 784) = 10.069; *p* < 0.001; η^2^*p* = 0.037) and the consumption of “healthy foods” (F (3, 784) = 12.593; *p* < 0.001; η^2^*p* = 0.046). This was not the case for consumption of “unhealthy foods” (F (3,784) = 2.485; *p* = 0.060; η^2^*p* = 0.009).

In the model corresponding to the MEDAS scores, post-hoc pairwise comparisons using the Bonferroni method found statistically significant differences between those who perceived their health to be excellent and the rest (with *p* < 0.001, *p* < 0.001 and *p* = 0.012 when compared with the good, passable, and poor perceived health group, respectively) indicating greater adherence to the MedDiet among those who perceived their health as excellent.

Concerning the model corresponding to the “healthy food” component, the pairwise comparisons showed statistically significant differences both for the group that perceived their health as excellent with respect to the rest (*p* = 0.001, *p* < 0.001 and *p* < 0.001 for good, passable and poor perceived health, respectively) and for those who indicated their health as good with respect to the rest (*p* = 0.026 and *p* = 0.004, for passable and poor, respectively)—indicating a relationship between higher consumption of fruits and vegetables and better-perceived health.

### 3.5. Mediation Analysis

In the case of SOC, its relationship with MEDAS scores was fully mediated by the consumption of “healthy foods” (standardized indirect effect = 0.059; 95% CI [0.017, 0.101]; Figure 1A).

The relationship between MEDAS scores and self-esteem was fully mediated by both the consumption of “healthy foods” (standardized indirect effect = 0.050; 95% CI [0.006, 0.093]) and the consumption of “unhealthy foods” (standardized indirect effect = −0.050; 95% CI [−0.083, −0.019]; Figure 1B), the effect of “healthy foods” is significantly greater than that of “unhealthy foods” (standardized indirect effect = 0.100; 95% CI [0.047, 0.155]).

The relationship between MEDAS scores and curiosity was partially mediated by reducing its total effect (*c*) from 0.156 (*p* < 0.001) to a direct effect (*c’*) of 0.137 (*p* = 0.001), which was a 12.18% reduction. This was mediated by both the consumption of “healthy foods” (positively, with standardized indirect effect = 0.065; 95% CI [0.025, 0.104]) and the consumption of “unhealthy foods” (negatively, with a standardized indirect effect = −0.045; 95% CI [−0.079, −0.013]), with the effect of the first mediator being significantly than the latter (standardized indirect effect = 0.110; 95% CI [0.058, 0.161]; Figure 1C).

The relationship with SWL was fully mediated by both “healthy food” consumption (standardized indirect effect = 0.047; 95% CI [0.004, 0.092]) correlating positively, and “unhealthy food” consumption (standardized indirect effect = −0.050; 95% CI [−0.083, −0.019]) correlating negatively. The contrast between both indirect effects showed a greater relevance of the consumption of “healthy foods” with respect to the consumption of “unhealthy foods”, the differences being statistically significant (standardized indirect effect = 0.098; 95% CI [0.042, 0.155]; Figure 1D).

Finally, the relationship with perceived health was partially mediated by the consumption of “healthy foods” (standardized indirect effect = 0.077; 95% CI [0.033, 0.120]; Figure 1E), which correlated positively, reducing the total effect (*c*) of the MEDAS scores from 0.189 (*p* < 0.001) to a direct effect (*c’*) of 0.108 (*p* = 0.012), which was a reduction of 42.86%.

## 4. Discussion

Our analysis of the eating habits of students at the University of Huelva revealed that only 29.9% showed high adherence to the MedDiet, a lower percentage than that found in students at the University of Granada and Murcia with estimates between 43.4–77.6% [63,87,88] but similar to those of the University of Castilla La Mancha with 24% [62]. Concerning the international setting, considerable variability has been observed in the eating patterns of university students in Europe [89]. However, the results of the present study are similar to those obtained in countries of the Mediterranean basin, with percentages of 26.3% and 22.7% in Greek and Italian university students, respectively [90,91]. However, in this study, the average consumption of fruits and vegetables was higher than less healthy foods such as chips and salty snacks, sweets, soft drinks, and energy drinks.

Analyses of these results, and particularly the observed intake of fruits and vegetables in preference to other foods, are positive as the consumption of these foods has been pointed out as one of the most useful health indicators [92,93,94] and is positively related to health perception and quality of life [15,95,96] both in psychological and physical health terms [1,15,18,97,98].

Regarding the psychological constructs, we found that the population studied presents a medium-high level of self-esteem (37.68), which is in agreement with data from other studies of similar populations [33]. Regarding SWL, the scores are medium-high (25.07), being similar to those found in some studies [99] and slightly higher than those found in others [100,101]. Our participants showed mean scores in curiosity (33.62), that were slightly lower than those found by Kashdan and Steger [46]. Regarding SOC, we found mean-high scores (57.72), these numbers being slightly lower than those of other studies [102]. Finally, the perceived health of university students is mostly between good and excellent (85.1%), these results being similar to those of other studies [23,24].

The principal component analysis of the food inventory provided results that allowed us to identify two clear groups. On the one hand, there is the fruits and vegetables group, and on the other hand, the group of chips and salty snacks, sweets, and soft drinks or sweetened beverages. The first group coincides with what is identified in the literature as “healthy foods” versus the second group that includes “unhealthy foods” [14], and thus we have named the groups in this way. These categories allowed us to conduct analyses and comparisons that extend beyond those using the items independently.

Regarding the relationship between the study variables, the results indicated that students with greater adherence to the MedDiet and consumption of “healthy foods” had greater SWL, curiosity, self-esteem, and SOC. This positive relationship is consistent with some of the studies that have examined the possible link between adopting a healthy diet and personal well-being, concluding that eating healthy has an impact on improved perceptions of well-being [15,103,104], physical health, and psychological adjustment [1,18,104,105] in both genders, and SWL in women [106], with a lower incidence of depressive symptoms [18,19]. Moreover, a relationship between “unhealthy food” consumption and SWL was observed.

A detailed analysis of each of these relationships reveals that, with respect to SWL, some studies such as that of Zaragoza-Martí et al. [106] maintain that adherence to the MedDiet improves health, increasing perceived quality of life and therefore SWL, which is also the case for the consumption of fruits and vegetables [15,107]. This relationship is directly associated with the perception of physical and psychological health in men and women and SWL in women [106]. Likewise, fruit and vegetable consumption is associated with an increase in SWL [104].

Concerning curiosity, in the model developed by Kashdan et al. [71,108], this is defined as a trait associated with SWL [46], and that in relation to healthy food consumption, some studies such as that of Conner et al. [109] maintain that curiosity is associated with personality traits of openness to experience and extroversion, and these traits are, in turn, associated with higher consumption of fruits and vegetables. Furthermore, consistent with the model of Kashdan et al. [71,108], studies such as Conner et al. [110] confirm that young adults who eat more fruits and vegetables show higher scores of psychological well-being and curiosity, as also indicated by the results obtained in our study.

Concerning self-esteem, high levels in this variable correlated positively with adherence to the MedDiet and “healthy food” consumption. Some authors suggest that physical activity and the adoption of a healthy diet are related to the perception of quality of life and self-esteem [88], although in general, studies do not seem to find a direct relationship between the two variables. The trend of the results suggests a pattern according to which perception self-esteem is high in the university population. However, it appears that this is not closely related to food intake [33] but to body image and Body Mass Index (BMI). In particular, a person with positive body perception and a healthy BMI would be more likely to follow a healthy diet and so would develop a higher level of self-esteem compared with those with a lower body image and unhealthy BMI [28,111,112]. Mendelson and Romano [30] had already reported this lack of relationship, indicating that self-esteem is negatively related to obesity derived from an unhealthy diet. More recently, other studies continue to corroborate this notion and indicate that the degree of personal acceptance of body image determines the development of self-esteem [113], so that poor body image is associated with low self-esteem [28,31,114]. Some recent attempts to find a direct relationship between the two variables include, for example, the study by Crichton et al. [115], who have analyzed the direct relationship between the adoption of the MedDiet and cognitive functioning variables such as memory, along with stress, anxiety, self-esteem, health perception, and physical activity, finding only a positive relationship with health perception and physical activity, a negative relationship with anxiety, depression and stress traits, and an absence of a relationship with self-esteem. In the study by Caamaño-Navarrete et al. [116] an attempt was made to determine the association between lifestyles (physical activity and eating habits), self-esteem and quality of life, finding that self-esteem was associated with quality of life and that lifestyles mediated this relationship in a positive manner (physical activity and adherence to the MedDiet).

Finally, and in relation to SOC or people’s ability to cope with unfamiliar situations and to stay healthy [35], our results are supported by studies establishing that high SOC scores are associated with healthier eating patterns [37,38,39,117] and higher fruit and vegetable intake [40,118,119] and, in general, with a greater tendency to adopt healthy behaviors at any stage of life. In a similar vein, the study by Wainwright et al. [40,41] found that people with a high SOC are less likely to be smokers, sedentary, and more likely to consume more fruits, vegetables, and fiber than those who had a low SOC, with the former being 20% less likely to die from any cause than the latter.

With respect to the psychological constructs analyzed, our results indicate that the interrelationships are high. In particular, these data suggest that the students’ perception of well-being has a positive impact on values of SWL, self-esteem, SOC, and curiosity. Various studies have shown the existence of positive correlations between these variables. For example, the study by Suraj and Singh [120] shows significant and positive relationships between SOC and psychosocial variables such as self-esteem, motivation, or achievement attribution; and negative relationships with affectivity, depression, psychological stress, and anxiety. In a similar vein, the study by García-Moya, Moreno, and Rivera [45] found a positive inter-relationship between variables such as SOC, health perception, and quality of life.

In relation to the study of potential mediating effects, our findings show how the consumption of “healthy foods” and “unhealthy foods” are relevant aspects in explaining the relationship between adherence to the MedDiet and self-esteem, curiosity, SWL, and perceived health.

In the case of self-esteem, curiosity, and SWL, we can observe how the consumption of “healthy foods” had a positive mediating effect. Conversely, the consumption of “unhealthy foods” has had a negative o null mediating effect. These results indicates the relationship between a higher adherence to the MedDiet and better psychological wellbeing and health perception would be explained because of the higher rate of “healthy foods” consumption and the lack of “unhealthy” one, which is in line with those of other studies that have identified the relationship between fruit and vegetable consumption and consumption of sweets and soft drinks and various psychological and health variables [104,110,117], along with studies that suggest the role of these food groups in the relationship between multiple psychological and health variables [41]. In addition, some studies have found that the consumption of certain MedDiet food groups is a predictor of increased in the adherence of MedDiet [62] and that the consumption of some components of the MedDiet was correlated with well-being [63]. All these studies, including ours, are compatible with the ideas proposed by García-Conesa et al. [57], pointing out the need to delve deeper into the role played by certain food groups of the MedDiet. The relevance of these results is based on the fact that the role played by the MedDiet in the relationship with other variables analyzed in this study is not exclusively due to the degree of follow-up of the MedDiet but depends to a large extent on the consumption of fruits, vegetables, fried potatoes, and salty snacks, sweets, and soft drinks or sugar-sweetened beverages.

On the other hand, for the relationship between adherence to the MedDiet and SOC and health perception, we found a statistically significant positive mediating effect only for consumption of “healthy foods”. This mediating effect in both relationships between fruit and vegetable consumption and SOC and health perception is in line with findings from previous studies [23,40,117,119,121], which would partially explain the relationships found between the MedDiet and perceived health [13].

## 5. Conclusions

Adherence to the MedDiet was low among students at the University of Huelva. However, consumption of fruits and vegetables was higher than that of other less healthy foods such as chips and salty snacks, sweets, soft drinks, and energy drinks.

The population studied showed medium-high scores in SWL, self-esteem, and SOC, medium scores in curiosity and mostly a good to excellent perception of health.

The food consumption inventory items can be grouped into two components: “healthy foods” (fruits and vegetables) and “unhealthy foods” (chips and salty snacks, sweets, and soft or sugary drinks).

Students with greater adherence to the MedDiet and consumption of “healthy foods” had higher SWL, curiosity, self-esteem, and SOC. However, a relationship was also found between the consumption of “unhealthy foods” and SWL.

A mediating effect of “healthy food” consumption on the relationships between MedDiet and SOC, self-esteem, curiosity, SWL, and perceived health was observed, whereas “unhealthy food” consumption negatively affected the relationship between MedDiet and self-esteem, curiosity SWL and perceived health.

## 6. Strengths and Limitations of the Study, and Future Lines of Research

To the best of our knowledge, this is the first study to assess several psychological constructs related to positive mental health together with adherence to the MedDiet and consumption of certain foods, as measured by independent instruments. Moreover, this is the first attempt to analyze possible indirect effects of these psychological constructs on perceived health and the MedDiet, finding that individual food groups play a mediating role rather than the general adoption of the MedDiet.

This study also has several limitations that must be acknowledged. One is that all the data obtained were collected through self-administered instruments. Therefore, certain associated biases may occur, although we attempted to mitigate this by making participation voluntary and informing the participants in advance of the confidentiality of the data treatment. In addition, due to the study’s cross-sectional design, it is not possible to establish causal relationships between the variables studied, which would require a longitudinal and experimental study that would allow us to verify the directionality and causality of the relationships explored. Finally, the sample can be considered representative of the selected university for this study and the Spanish university context, however, the distribution by sex or area of knowledge can be widely different in a different international university context.

Scientific evidence shows that adopting a healthy diet is positively associated with SWL, self-esteem, SOC, curiosity, and a better perception of health, so it would be of interest to use these data as a basis for developing training programs on eating habits in the university population. In addition, future lines of research should include new indicators that refer to other food groups, such as meats, seafood, or oil and other fats, and thus test relationships based on MedDiet and mental health.

## Figures and Tables

**Figure 1 nutrients-13-03769-f001:**
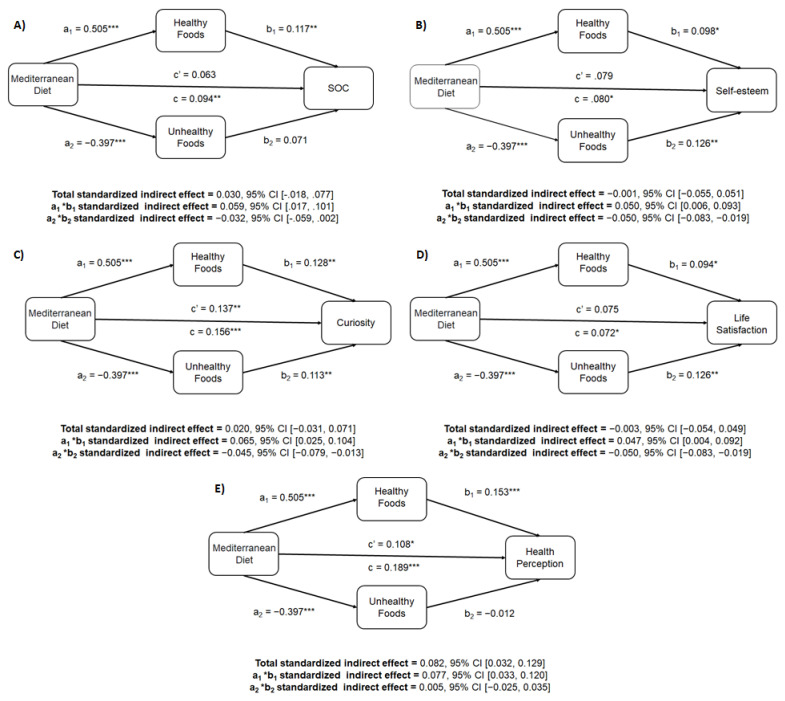
Mediation models of “healthy food” and “unhealthy food” consumption on (**A**) SOC, (**B**) self-esteem, (**C**) curiosity, (**D**) SWL and (**E**) perceived health. 95% CI: 95% Confidence Interval; * *p* < 0.05; ** *p* < 0.01; *** *p* < 0.001.

**Table 1 nutrients-13-03769-t001:** Frequency and percentage of food consumption.

	Rarely or Never	Once a Week	2–4 Times a Week	5–6 Times a Week	7 Times a Week or More
Fruits	138 (17.5%)	102 (12.9%)	284 (36%)	99 (12.6%)	165 (21%)
Chips and salty snacks	209 (26.6%)	208 (26.4%)	285 (36.2%)	65 (8.2%)	21 (2.6%)
Vegetables	51 (6.5%)	102 (12.9%)	305 (38.7%)	161 (20.4%)	169 (21.5%)
Sweets	227 (28.9%)	167 (21.2%)	230 (29.2%)	84 (10.7%)	80 (10.2%)
Energy drinks	671 (85.1%)	53 (6.7%)	40 (5.1%)	8 (1%)	16 (2%)
Soft drinks	357 (45.3%)	160 (20.3%)	157 (19.9%)	34 (4.3%)	80 (10.2%)

**Table 2 nutrients-13-03769-t002:** Eigenvalues of the extracted components.

Initial Eigenvalues			
Component	Total	Variance	Accumulated
1	2.103	35.047	35.047
2	1.185	19.747	54.794
3	0.954	15.907	
4	0.683	11.386	
5	0.571	9.514	
6	0.504	8.398	

**Table 3 nutrients-13-03769-t003:** Pattern matrix after Promax rotation.

Variable	Component 1	Component 2
Fruits		0.873
Potato chips and salty snacks	0.770	
Vegetables		0.808
Sweets (candies and chocolates)	0.741	
Energy drinks	0.532	
Soft drinks or beverages with sugar	0.626	

**Table 4 nutrients-13-03769-t004:** Correlation matrix.

	“Healthy Foods”	“Unhealthy Foods”	SWL	Curiosity	Self-Esteem	SOC
MEDAS Score	0.505 ***	−0.397 ***	0.072 *	0.156 ***	0.079 *	0.094 **
“Healthy foods”	-	−0.252 ***	0.100 **	0.169 ***	0.107 ***	0.131 ***
“Unhealthy foods”		-	0.073 *	0.027	0.070 ^+^	0.016
SWL		.	-	0.284 ***	0.561 ***	0.503 ***
Curiosity				-	0.368 ***	0.213 ***
Self-esteem					-	0.638 ***

MEDAS, Mediterranean Diet Adherence Screener; SWL, Satisfaction with Life; SOC, Sense of Coherence. ^+^ = 0.05 * *p* < 0.05 ** *p* < 0.01 *** *p* < 0.001.

## Data Availability

The data that support the findings of this study are available upon request from the authors.

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
