# Peer review of "Mediterranean Diet, Psychological Adjustment and Health Perception in University Students: The Mediating Effect of Healthy and Unhealthy Food Groups"

_nutrients, 2021, doi:10.3390/nu13113769_

Round 1
Reviewer 1 Report
This is a very interesting paper in which authors try to identify the relationships between eating habits and psychological adjustment and health perception, and to analyze potential mediating role of healthy and unhealthy foods in the relationship between adherence to the Mediterranean diet and the psychological constructs and health perception. Overall it is a well structured paper and addresses an important issue not only substantial and relevant for clinical placements and physicians but also for the fields of primary prevention. However, I have some minor revisions to suggest:
1) p. 2, line 56. Please replace "DM" with "MD"
p. 6, line 282. Please change "DM" with "MD"
2) I believe that the fact that the 75% of the participants were females and only 25% males should be added in the limitation section.
3) Don't you believe that there is an extra bias from the participants who belong to the "Health Science" education group?
4) Statistical analysis section: Please identify "the nature of each variable". (When mean ± Standard deviation and when the frequencies were used ?)
5) Table 1: Please repalce "Little or nothing" with "Rarely or Never".
Reviewer 2 Report
Thank you for the opportunity to review this manuscript titled Mediterranean diet, psychological adjustment and health perception in university students: the mediating effect of healthy and unhealthy food groups by Velez-Toral et al. The findings presented are indeed of interest to this journals readership. However, before this paper can be accepted I have a few queries presented below, in no particular order of importance. Lastly, could the authors also present relevant results in the abstract, rather than a description of the findings.
Introduction
Line 42: should be low intake of saturated fats given that lipid consumption (MUFA in particular) is generally quite high with adherence to a traditional MedDiet. Please change to saturated fat.
Line 56: what is DM? Do you mean MD – Mediterranean diet? On this point, I would consider changing the abbreviation of the Mediterranean diet from MD to MedDiet, which is a more accepted abbreviation in the research literature.
There are also other potential barriers (e.g. cost, socioeconomic status, knowledge, cooking skills etc) which have consistently been cited in the literature related to poor adherence to a MedDiet (or food groups consistent with a MedDiet, e.g. fruit and vegetable intake). These are also pertinent to a university population. Such factors should also be addressed in the introduction.
Methods:
Lines 143-153: This appears to be results i.e. no. of respondents completing the survey, student discipline area etc. I wonder if this would be better presented in the results section of the manuscript?
In the description of the MEDAS, I don’t think the term ‘does not satisfy the condition or satisfies the condition’ is an accurate reflection of the questionnaire. Please consider the following ‘does not meet criteria or meets criteria’.
Line 163: This is not quite accurate. The final two questions of the MEDAS are related to dietary habits/characteristics consistent with a traditional MedDiet – for example, preferential consumption of white over red meat and frequency of consumption related to pasta/rice/vegetables cooked with a sauce of tomato, onion, garlic, leeks sauteed in olive oil (Sofrito). Please amend.
It is unclear to me what the life satisfaction and curiosity scales actually measure? For example, what items/domains do each of these tools assess. The reporting of self-esteem and sense of coherence on the other hand were clearer. This description should be teased out further in the methods.
Results:
I would suggest that a mean score of 7.42 for the MEDAS is actually reflective of moderate adherence to the MedDiet, rather than low adherence. Please amend throughout the manuscript.
Please consider identifying abbreviations as footnotes under Table 4: e.g. SWL, SOC etc
Discussion:
Given that >40% of the cohort were from a Health Sciences background, the authors should acknowledge the potential confounder here given that these students are more likely to engage in healthier behaviours such as increased fruit and vegetable consumption.
